# AP2 hemicomplexes contribute independently to synaptic vesicle endocytosis

Mingyu Gu[1], Qiang Liu[1], Shigeki Watanabe[1], Lin Sun[2], Gunther Hollopeter[1], Barth D Grant[2], Erik M Jorgensen[1,3]*

[1]Department of Biology, Howard Hughes Medical Institute, University of Utah, Salt Lake City, United States; [2]Department of Molecular Biology and Biochemistry, Rutgers University, Piscataway, United States; [3]Department of Biology, University of Utah, Salt Lake City, United States

**Abstract** The clathrin adaptor complex AP2 is thought to be an obligate heterotetramer. We identify null mutations in the α subunit of AP2 in the nematode *Caenorhabditis elegans*. α-adaptin mutants are viable and the remaining μ2/β hemicomplex retains some function. Conversely, in μ2 mutants, the alpha/sigma2 hemicomplex is localized and is partially functional. α-μ2 double mutants disrupt both halves of the complex and are lethal. The lethality can be rescued by expression of AP2 components in the skin, which allowed us to evaluate the requirement for AP2 subunits at synapses. Mutations in either α or μ2 subunits alone reduce the number of synaptic vesicles by about 30%; however, simultaneous loss of both α and μ2 subunits leads to a 70% reduction in synaptic vesicles and the presence of large vacuoles. These data suggest that AP2 may function as two partially independent hemicomplexes.

*For correspondence:
jorgensen@biology.utah.edu

**Competing interests:** The authors declare that no competing interests exist

## Introduction

Proteins on the surface of cells are removed from the plasma membrane by endocytosis. Many cargo proteins are recruited to sites of endocytosis by the tetrameric adaptor complex AP2 (*Mahaffey et al., 1990*; *Traub, 2003*). The adaptor complex in turn recruits the coat protein clathrin to the membrane. Clathrin converts the raft of cargo and adaptor proteins into a budding vesicle by forming a scaffold that shapes the membrane (*Mashl and Bruinsma, 1998*). Clathrin-mediated endocytosis probably functions in all tissues, but it is unclear whether this process is suited to the particularly high rates of endocytosis required at nerve terminals. Nonetheless, the predominant mechanism for synaptic vesicle endocytosis is thought to be mediated via AP2 and clathrin (*Dittman and Ryan, 2009*). Testing this model by disrupting clathrin is difficult to interpret because trafficking from the trans-Golgi relies in part on clathrin-coated vesicles. Thus, genetic analysis of AP2 mutants is more specific for endocytic trafficking of proteins from the cell surface.

The AP2 adaptin complex has four subunits—two large subunits α and β2, a medium subunit μ2, and a small subunit σ2 (*Matsui and Kirchhausen, 1990*). It is generally thought that adaptor complexes act as obligate tetramers; loss of one subunit will destabilize the entire complex (*Dell'Angelica et al., 1998*; *Kantheti et al., 1998*; *Collins et al., 2002*; *Motley et al., 2003*; *Traub, 2003*; *Nakatsu et al., 2004*; *Mitsunari et al., 2005*; *Kim and Ryan, 2009*). AP2 functions at the plasma membrane as an interaction hub for transmembrane cargoes, accessory proteins, and clathrin (*Traub, 2003*; *Robinson, 2004*). Loss of single AP2 subunits is known to disrupt endocytosis at the plasma membrane. A null allele in α-adaptin is lethal and leads to an absence of synaptic vesicles at neuromuscular junctions in *Drosophila* embryos and thus appears to disrupt endocytosis (*Gonzalez-Gaitan and*

**eLife digest** The cell membrane is a busy place, with cell-surface proteins continually added and removed according to the needs of the cell. Each protein extends a polypeptide tail into the cell cytoplasm. When a protein is to be removed from the cell surface, its tail recruits a protein complex known as the AP2 adaptor to the membrane. AP2 then recruits a coat protein called clathrin, which forms a spherical scaffold around the adaptor, the target protein and the surrounding membrane, enclosing them inside a vesicle that breaks off from the membrane and enters the cell.

Endocytosis is particularly common in neurons, which use it as a means of recycling proteins at synapses—the contact points between nerve cells. However, it is unclear whether synaptic-vesicle recycling also involves clathrin and AP2. To address this question, Gu et al. examined mutant nematode worms (*C. elegans*) in which the composition of AP2 had been altered.

AP2 has four subunits, called α, β2, μ2 and σ2, and Gu et al. produced worms that lack either the α- or μ2-subunit, or both. Few worms that lacked both subunits survived. Surprisingly, however, worms that lacked just one subunit were viable, despite previous evidence that AP2 requires all four subunits to be functional. Nevertheless, these single mutants produced 30% fewer synaptic vesicles compared to wild-type worms. To examine the consequences of both subunits being absent, Gu et al. rescued the double mutants by selectively expressing AP2 in their skin. These animals—which still lack AP2 in their nervous systems—produced 70% fewer synaptic vesicles than their wild-type counterparts.

The results show that AP2 does not need all four of its subunits and that it can exist as two semi-independent hemicomplexes. Moreover, Gu et al. show that *C. elegans* uses at least two endocytotic mechanisms (AP2-dependent and independent) to recycle vesicles and so maintain synaptic function.

*Jackle, 1997*). Similarly, in *C. elegans* loss of either of α- or β-adaptin by RNA interference perturbs the endocytosis of yolk protein from the plasma membrane (*Grant and Hirsh, 1999*). These data suggest that loss of either large subunit eliminates AP2 function.

Recent data suggest that the medium subunit μ2 may not play an essential role for the endocytosis of synaptic vesicle components from the plasma membrane (*Gu et al., 2008*; *Kim and Ryan, 2009*). Although μ2 is required in part for the localization of clathrin at synapses (*Gu et al., 2008*), synaptic vesicles and constituent proteins are still recycled in the absence of μ2 (*Gu et al., 2008*; *Kim and Ryan, 2009*). These contrasting results for α-adaptin vs μ2-adaptin mutants from different organisms suggest that α-adaptin is essential for synaptic vesicle endocytosis, whereas the μ2 subunit may not be essential.

Here we evaluate the function of AP2 at synapses by studying mutations in α- and μ2-adaptins in *C. elegans.* Because null mutations for both of these genes are viable, we can compare the loss of these AP2 subunits in a single organism for the first time. Mutants lacking α-adaptin retain a partially functional AP2 hemicomplex consisting of μ2 and β-adaptin. Mutants lacking both α and μ2 subunits exhibit a more severe phenotype than the single mutants and are subviable. These results suggest that the single subunits retain some function, but that the double mutants lack all AP2 function. Nevertheless a moderate level of synaptic transmission remains in the double mutant and is able to sustain locomotory behavior, suggesting the presence of an AP2 independent mechanism capable of maintaining synaptic transmission at the synapse.

## Results

### α-adaptin mutations

In *C. elegans*, α-adaptin is encoded by the *apa-2* gene (*Figure 1A*). Two alleles of *apa-2* have been isolated (*Figure 1B*): *b1044* is a 925 bp deletion that starts within the second intron, extends to the fourth exon and deletes a large fraction of the trunk domain. *ox422* is premature stop mutation at Lys215 and would lead to a truncation of α-adaptin from the middle of the trunk domain to the carboxy terminus including the ear domain (*Figure 1C*). We did not detect full-length APA-2 protein from either of these alleles (*Figure 1—figure supplement 1*), and they are likely to be null mutations.

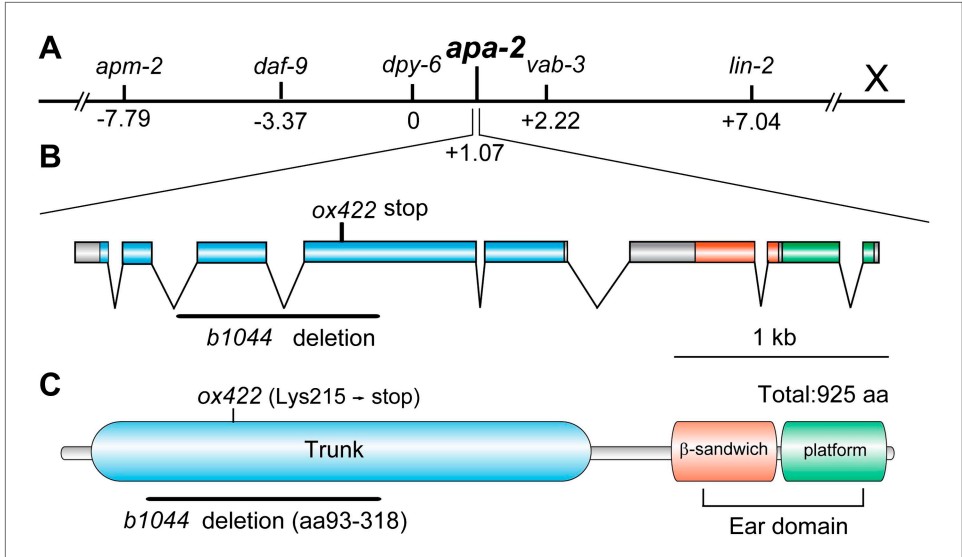

**Figure 1**. *apa-2* cloning. (**A**) Genetic map position of *apa-2* on chromosome X. (**B**) Genomic structure of the *apa-2* gene and the nature of mutant alleles. *b1044* is a 925 bp deletion from the second intron to the fourth exon. *ox422* is an A to T transversion. (**C**) Protein domain structure of alpha adaptin. *b1044* causes a deletion of aa93-318 in the trunk domain. *ox422* changes Lys215 to a premature stop.

The following figure supplements are available for figure 1:

**Figure supplement 1**. Western blot of α adaptin mutants *apa-2(ox422)* and *apa-2(b1044)*.

To determine the expression pattern of *apa-2*, we inserted the coding sequence for GFP in frame at the 3′ end of the open reading frame (*Figure 2A*). The GFP fusion construct rescued the mutant phenotype (data not shown). The *apa-2* gene appears to be expressed in most cells of the animal, and is highly expressed in the nervous system (*Figure 2B,C*).

Unlike α-adaptin mutants in *Drosophila* (*Gonzalez-Gaitan and Jackle, 1997*), worms missing α-adaptin in *C. elegans* are viable; they grow to adulthood and are grossly similar to μ2-adaptin *(apm-2)* mutants (*Figure 3A*). About 30% of α mutants (*Figure 3A*) and about 5% of μ2 mutants (*Gu et al., 2008*) have cuticle protrusions on either side of the head called 'jowls'. However, the variable dumpy phenotype of *apa-2* is less severe than that of *apm-2* (*Figure 3B*). The α-adaptin mutants are egg-laying defective and mildly uncoordinated; they crawl forward well but are slightly jerky as they move backward. *apa-2* mutants exhibit only mild defects in thrashing when placed in liquid (*Figure 3C*).

Expression of APA-2::GFP under a ubiquitous promoter can fully rescue the mutant phenotypes including the cuticle protrusions (*Figure 3A–C*; *Figure 3—figure supplement 1*). Expression of the α-adaptin specifically in the epidermis (the equivalent of skin in *C. elegans*) rescues the cuticle pheno-type (*Figure 3A*; *Figure 3—figure supplement 1*), which is similar to rescue experiments in μ2-adaptin mutants (*Gu et al., 2008*). However unlike μ2-adaptin mutants, the dumpy phenotype of *apa-2* is res-cued by neuron-specific but not skin-specific expression (*Figure 3B*). In fact, the neuron-rescued worms are longer than the wild type (*Figure 3B*). Thus, α- and μ2 mutants exhibit similar but distinct pheno-types suggesting α- and μ2-adaptins may not be required for identical functions of AP2 in worms.

If the functions of α-adaptin and μ2-adaptin are different, then the double mutants will be synthetic, that is the phenotype of the double mutant will be much more severe than the single mutants. Indeed, when the *apa-2* and *apm-2* mutations are combined, only 4.3% of the double mutants coming from a heterozygote are viable and the brood size of these survivors is reduced to 1.4% compared to the wild type (*Figure 4A,B*). The rare escapers grow twice as slowly as wild-type animals and are sick and dumpy (*Figure 4C* and *Figure 4—figure supplement 1*). RNAi of μ2 in α mutants and α in μ2 mutants produced similar results (data not shown). These data suggest residual function of AP2 remains in both *apa-2* and *apm-2* single mutants.

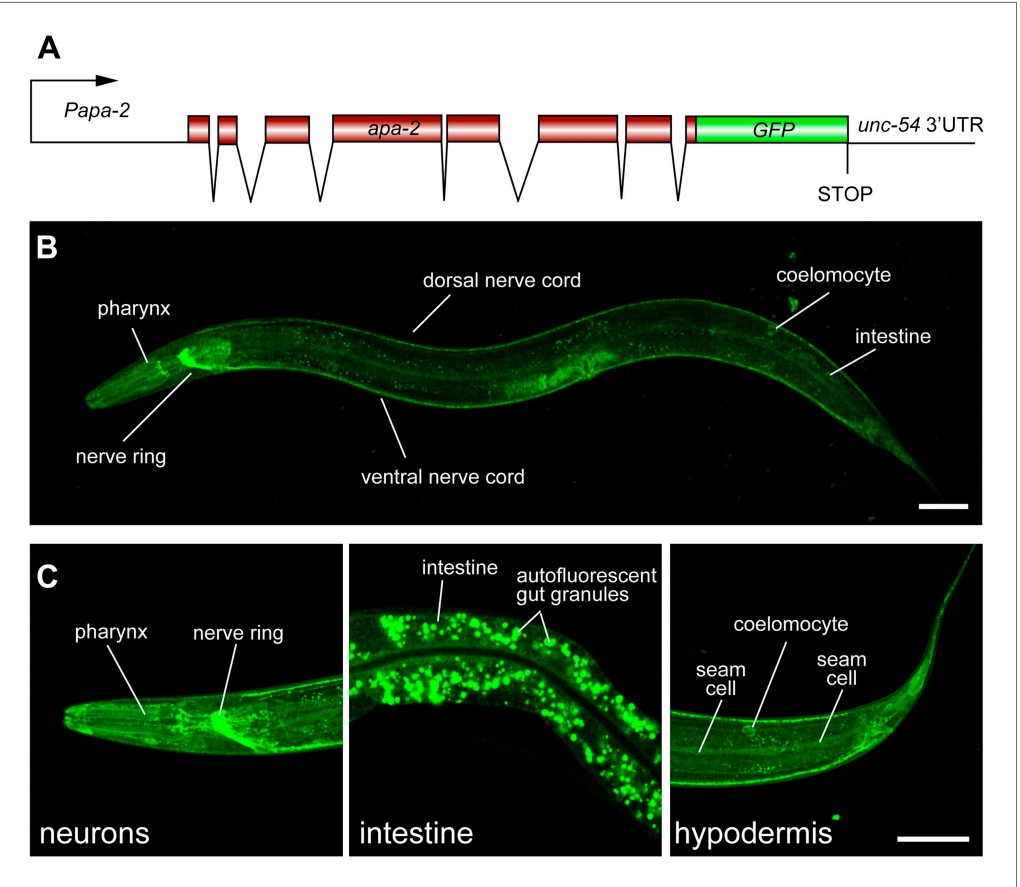

**Figure 2**. α-adaptin is expressed ubiquitously. (**A**) Schematic of *apa-2::GFP* translational reporter construct. The APA-2::GFP fusion construct is expressed under the control of the *apa-2* promoter (1.9 kb upstream of ATG) from an extrachromosomal array in a *lin-15* rescued background. (**B**) The expression pattern of the translational fusion protein APA-2::GFP in young adult hermaphrodite. The worm is oriented anterior left and dorsal up. GFP fluorescence is observed ubiquitously in transgenic worms. (**C**) Detailed images of APA-2::GFP expression in three tissues: nervous system, intestine and hypodermis. The scale bar represents 50 μm.

## Cargo specific defects in α-adaptin vs μ2-adaptin mutants

At a cellular level, α-adaptin mutants exhibit defects in endocytosis. Yolk is a lipoprotein particle composed of lipids and lipid-transport proteins called vitellogenins. Yolk particles are synthesized and secreted by the intestine and are then taken up from the extracellular space by maturing oocytes via receptor-mediated endocytosis. Yolk endocytosis is clathrin-dependent and can be assayed in animals expressing GFP-tagged vitellogenin-2 (YP170::GFP) (*Grant and Hirsh, 1999*; *Sato et al., 2009*). In wild-type worms, YP170::GFP is enriched in the three most mature oocytes near the spermatheca. In α-adaptin mutants, the number of GFP-positive oocytes is decreased to one or two cells (*Figure 5A,B*), which is similar to the defect in μ2 mutants (*Gu et al., 2008*). By contrast, strong defects in YP170::GFP endocytosis are observed in mutants lacking the alternative clathrin adaptor Disabled (*Holmes et al., 2007*). Thus, the AP2 complex appears to assist Disabled for yolk endocytosis, and α- and μ2-adaptin do not contribute differentially to this process. To determine if α- and μ2-adaptin contribute differentially to endocytosis, we needed to identify specific cargo.

There is no known α-adaptin specific cargo in *C. elegans*. However, the α-adaptin subunit is involved in binding cargo with di-leucine motifs (*Kelly et al., 2008*). We constructed an artificial cargo protein known to bind α-adaptin (*Figure 5C*). We tagged the human CD4 protein with GFP, and appended the di-leucine motif (ExxxLL) from HIV Nef to the carboxy terminus (*Doray et al., 2007*). We expressed the construct in the intestine. In wild-type worms CD4-dileucine is localized to intracellular compartments;

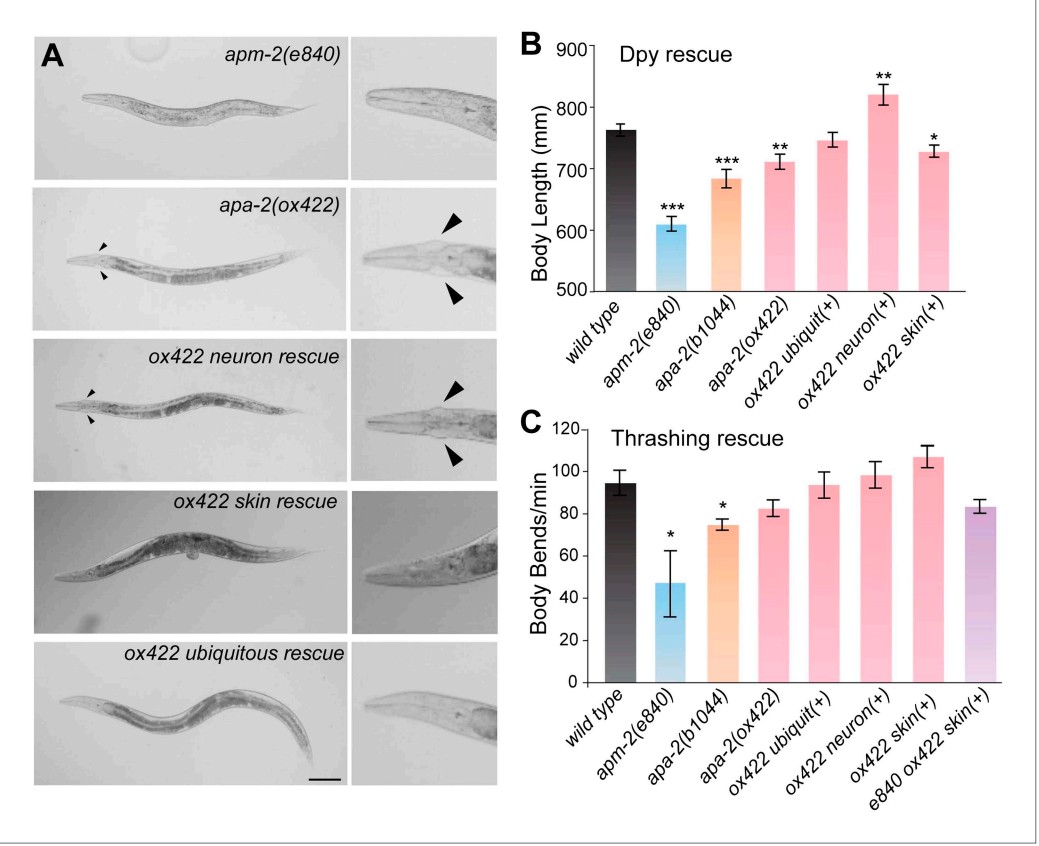

**Figure 3**. Tissue-specific rescue of α-adaptin mutant. (**A**) Bright field images *apm-2(e840)* and tissue-specific rescue of *apa-2(ox422)* mutants. Worms are rescued by strains carrying single-copy transgenes. The jowls are indicated by the black arrowheads; most *apm-2* animals lack jowls. The scale bar represents 100 μm. (**B**) The dumpy phenotype of *apa-2* mutants is rescued by neuronal expression. Body length of *apm-2(e840)* (deletion allele of μ2 adaptin), *apa-2* mutants and *apa-2* tissue-specific rescued animals. Average body length at the L4 stage in μm ± SEM: wild type 763 ± 10, *apm-2(e840)* 609 ± 12 (p<0.0001), *apa-2(ox422)* 711 ± 12 (p=0.0037), *apa-2(b1044)* 684 ± 15 (p<0.0001), skin-rescued *apa-2(ox422)* 727 ± 10 (p=0.0203), neuron-rescued *apa-2(ox422)* 820 ± 17 (p=0.0098), ubiquitous rescued *apa-2(ox422)* 773 ± 12 (p=0.5301). n = 10 L4 worms. (**C**) Locomotion assay. Average body bends per minute ± SEM: wild type 94.4 ± 6.0, *apm-2(e840)* 46.0 ± 17.4 (p=0.0478), *apa-2(ox422)* 82.4 ± 4.0 (p=0.1347), *apa-2(b1044)* 74.6 ± 2.7 (p=0.0168), ubiquitously-rescued *apa-2(ox422)* 93.4 ± 6.2 (p=0.9106), neuron-rescued *apa-2(ox422)* 98.2 ± 6.3 (p=0.6738), skin-rescued *apa-2(ox422)* 106.8 ± 5.2 (p=0.1570), skin-rescued *apa-2(ox422) apm-2(e840)* 83.2 ± 3.2 (p=0.1382). n = 5 adult hermaphrodites. n of *apm-2* = 7. * p<0.05, ** p<0.01, *** p<0.001.

The following figure supplements are available for figure 3:

**Figure supplement 1**. Tissue-specific expression of APA-2::GFP.

however, in *apa-2* mutants CD4-dileucine accumulates abnormally on basolateral membranes (*Figure 5D*). By contrast in *apm-2* mutants, the CD4-dileucine accumulation on the plasma membrane is milder (69% compared to α mutants; *Figure 5D,E*). These data suggest that recovery of di-leucine cargo depends on α-adaptin more than μ2-adaptin.

MIG-14/wntless is a μ2-dependent cargo (*Pan et al., 2008*). In the absence of the μ2 subunit, MIG-14::GFP is strongly mislocalized to the basal and lateral surfaces of intestine cells (*Figure 5F,G*). In the absence of α-adaptin, MIG-14::GFP is only weakly mislocalized on the basolateral surface (27% compared to μ2 mutants; *Figure 5F,G*). MIG-14 contains a tyrosine in its carboxy terminus (μ2 consensus target is Yxxφ), but it is not known if this sequence is required for μ2 binding. As a control, the endocytosis of clathrin-independent cargo hTAC is unaffected in both adaptin mutants (*Figure 5—figure supplement 1, 2*). Taken together, these data suggest that partially functional AP2 complexes might be present in mutations that eliminate single subunits.

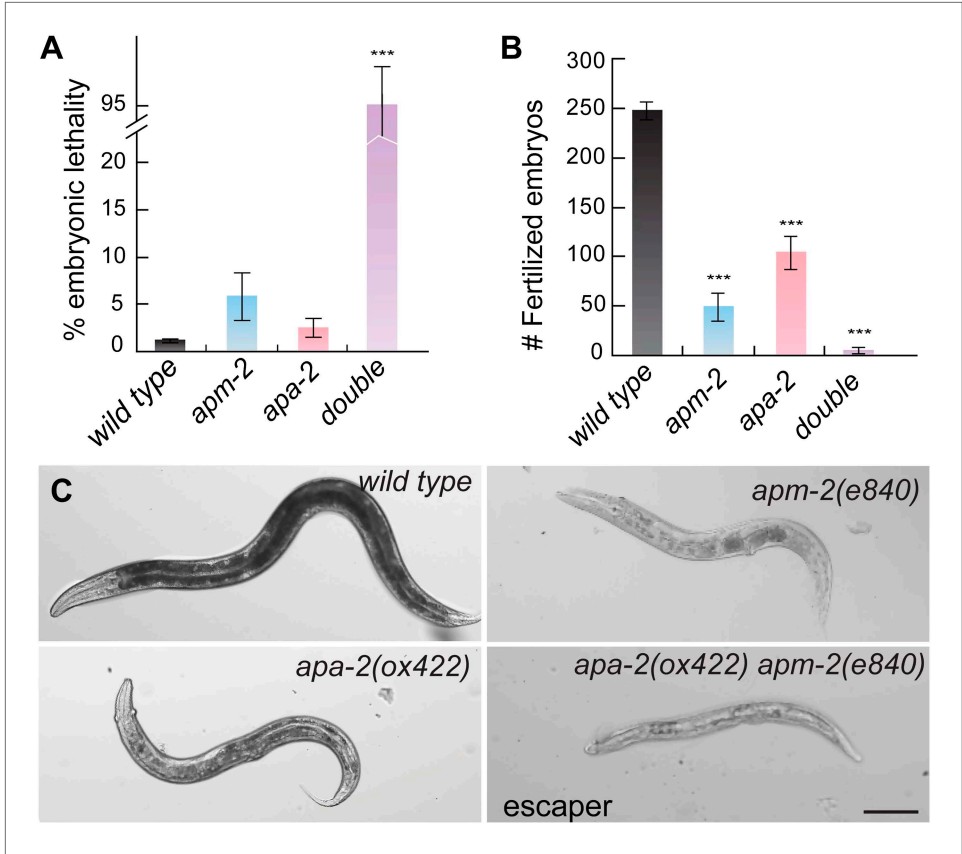

**Figure 4**. α- and μ2-adaptin double mutant is synthetic. (**A**) Embryonic lethality (% total embryos) of AP2 mutants ± SEM: wild type 1.13 ± 0.25 n = 10, *apa-2(ox422)* 2.42 ± 0.96 n = 9 (p=0.1902), *apm-2(e840)* 5.85 ± 2.56 n = 10 (p=0.0831), *apa-2(ox422) apm-2(e840)* 95.70 ± 4.30 n = 11 (p<0.0001). *** p<0.001. (**B**) The brood size of AP2 mutants±SEM: wild type 247.3 ± 8.8 n = 10, *apa-2(ox422)* 104.8 ± 16.9 n = 9 (p<0.0001), *apm-2(e840)* 49.0 ± 14.2 n = 10 (p<0.0001), *apa-2(ox422) apm-2(e840)* 3.4 + 3.1 n = 11 (p<0.0001). (**C**) Bright-field images of the wild type, *apm-2(e840), apa-2(ox422)* and a surviving *apa-2(ox422) apm-2(e840)* adult. The scale bar represents 100 μm.

The following figure supplements are available for figure 4:

**Figure supplement 1**. AP2 mutants exhibit slowed postembryonic development.

## AP2 hemicomplexes

The open form of AP2 can be considered as two hemicomplexes: the α and the σ2 subunits are in close contact, and the μ2 and β subunits are in close contact in both open and closed forms of the complex (*Collins et al., 2002*; *Jackson et al., 2010*). However, these two hemicomplexes are only loosely associated in the open form of the AP2 complex (*Jackson et al., 2010*). Here we demonstrate that in the absence of α-adaptin that a μ2-β hemicomplex remains, and that in the absence of μ2-adaptin that a α-σ2 complex remains in vivo.

The β- and μ2-adaptins, which are not closely associated with α-adaptin, are stable in the absence of α−adaptin. Transgenes expressing GFP-tagged AP2 subunits were inserted as single copy transgenes and crossed into *apa-2* mutants. Tagged β-adaptin and μ2-adaptin are localized to synaptic regions of the nerve ring (the major neuropil of the worm, *Figure 6B,C*) and at the plasma membrane in oocytes (*Figure 6—figure supplement 1*). The level of μ2-adaptin is reduced to 40% in *apa-2* mutants as assayed by western blot (*Figure 6—figure supplement 1*) or 20% as measured by fluorescence (*Figure 6*). On the other hand, the small σ2 subunit, which is normally tightly bound to α-adaptin, is unstable in *apa-2* mutants. Tagged σ2 is no longer detectable in the nerve ring (*Figure 6*) or in maturing oocytes (*Figure 6—figure supplement 2*), and the protein is reduced to about 10% of the wild-type level as assayed by fluorescence (*Figure 6*, *Table 1*) or western blot (*Figure 6—figure supplement 2*).

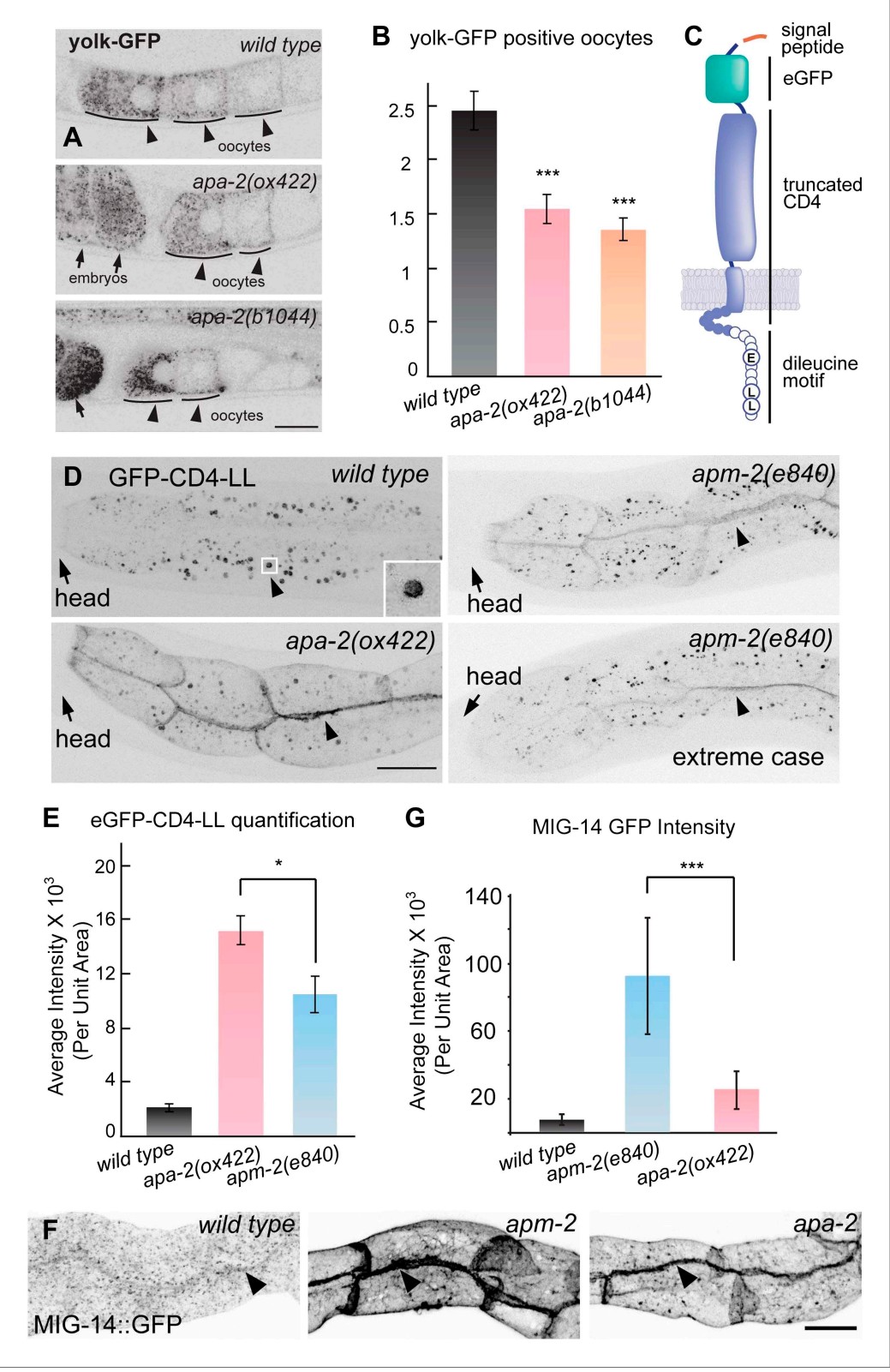

**Figure 5.** Endocytosis of α- and μ2-adaptin-dependent cargo. Fluorescence images have been inverted to aid visualization of signals. (**A**) Yolk protein, YP170::GFP is endocytosed by maturing oocytes in the wild type and both *apa-2* mutants. Black arrowheads point to maturing oocytes and black arrows point to fertilized embryos. (**B**) The number of YP170::GFP positive oocytes ± SEM: wild type 2.5 ± 0.2, *apa-2(ox422)* 1.6 ± 0.1 (p=0.0003), *apa-2(b1044)*

*Figure 5. Continued on next page*

*Figure 5. Continued*

1.4 ± 0.1 (p<0.0001). n = 20 adult hermaphrodites, *** p<0.001. Two-tailed Student's *t*-test. (**C**) A diagram of eGFP-CD4 artificial cargo. eGFP was flanked by two 12 aa flexible linkers and inserted after the secretion signal peptide, the extracellular domain of CD4 was truncated to include one immunoglobulin domain, the cytoplasmic domain of CD4 was removed leaving a seven aa tail (**Feinberg et al., 2008**), and the 11 aa Nef di-leucine motif was fused after CD4 (**Doray et al., 2007**). Circles represent amino acids on the cytoplasmic face. (**D**) eGFP-CD4-LL localization in intestine in the wild type and α adaptin mutant *(apa-2(ox422) X)* and μ2 adaptin mutant *(apm-2(e840) X)* mutants. Black arrowheads point to the intracellular organelle in the wild type (see inset) and to the lateral surface of the plasma membrane in the mutants. (**E**) Quantification of fluorescence intensity of eGFP-CD4-LL in wild type and AP2 mutants. Total fluorescence was measured from regions of interest defined on the basolateral membrane and averaged. Fluorescence intensity arbitrary units mean ± SEM: wild type 1998 ± 275 n = 5, *apa-2(ox422)* 14,907 ± 990 n = 6 (p<0.0001), *apm-2(e840)* 10,310 ± 1342 n = 6 (p=0.0004). The p value between *apa-2* and *apm-2* is <0.0203. * p<0.05. (**F**) Endocytosis of MIG-14/wntless in the intestine in the wild type, μ2 adaptin mutant *(apm-2(e840) X)* and an α adaptin mutant *(apa-2(ox422) X)*. Fluorescence images are inverted to better view dim GFP fluorescence. (**G**) Quantification of fluorescence intensity of MIG-14 in wild type and AP2 mutants. Total fluorescence was measured from regions of interest defined on the basolateral membrane and averaged. The data were captured on a Zeiss LSM 510 and the spectral fingerprinting feature was used to remove intestinal autofluorescence. Fluorescence intensity arbitrary units mean ± SD: wild type 7363 ± 3498 n = 18, *apm-2(e840)* 92,648 ± 34,237 n = 18 (p<0.0001), *apa-2(ox422)* 25,110 ± 11,570 n = 18 (p<0.0001). The p value between *apa-2* and *apm-2* is <0.0001. *** p<0.001.

The following figure supplements are available for figure 5:

**Figure supplement 1**. An AP2-independent cargo is not affected by AP2 subunits mutants.

**Figure supplement 2**. Quantification of total fluorescence intensity of hTAC in the wild type and AP2 mutants.

Conversely, α-adaptin and σ2-adaptin are localized in the absence of μ2-adaptin. In *apm-2* mutants, tagged α-adaptin is still localized to the synapse (**Figure 6D**) and the plasma membrane of coelomocytes. α-adaptin levels are only reduced to 60% as assayed by western blot (**Gu et al., 2008**) or 40% as assayed by fluorescence. Tagged σ2-adaptin is still localized to the nerve ring (**Figure 6A**) and is reduced to 40% as assayed by fluorescence (**Figure 6**, **Table 1**). On the other hand, the large β subunit, which is normally tightly bound to μ2-adaptin, is unstable in *apm-2* mutants. The β subunit is shared by AP1 and AP2 in *C. elegans*, and tagged β subunit fluorescence is visible in cell bodies in *apm-2* mutants. However, tagged β is no longer detectable in the synapse-rich region of the nerve ring in the absence of μ2-adaptin (**Figure 6B**, **Table 1**). Taken together, these data suggest that AP2 hemicomplexes are partially stable and can function in vivo in the absence of a complete AP2 complex.

## Synaptic vesicle biogenesis is defective in α and α-μ2 double mutants

To study the function of AP2 components in neurons, we rescued the mutant defects in the epidermis. Providing AP2 function in the skin was necessary for two reasons: First, AP2 components are required in the epidermis to play non-autonomous roles in synaptic development (**Gu et al., 2008**; **Pan et al., 2008**). Second, due to the low viability of the double mutant, it is impossible to maintain as a homozygous strain. However, when α- and μ2-adaptins are simultaneously introduced back into the epidermis, 100% of the double mutants grow to adults. The skin-rescued worms have no detectable APA-2::GFP in the nervous system (**Figure 3—figure supplement 1**) and the skin promoter P*dpy-7* is only expressed in larval stages during development (**Johnstone and Barry, 1996**). These rescued animals are still egg-laying defective and slow-growing, but they provide an opportunity to study synaptic vesicle endocytosis in AP2-deficient synapses.

We assayed the synaptic localization of α-adaptin by expressing an *apa-2::GFP* fusion construct specifically in GABA neurons. α-adaptin colocalizes with a synaptic vesicle protein, synaptobrevin, in both the dorsal and ventral nerve cords (**Figure 7—figure supplement 1**). This result suggests that α-adaptin associates with synaptic varicosities, similar to μ2-adaptin (**Gu et al., 2008**).

We examined the requirement of AP2 for the recycling of several synaptic-vesicle proteins. In *C. elegans* mutants lacking particular adaptor proteins, the cognate cargo protein diffuses into axons. For example in AP180 mutants, synaptobrevin is no longer concentrated at synapses but is diffuse in

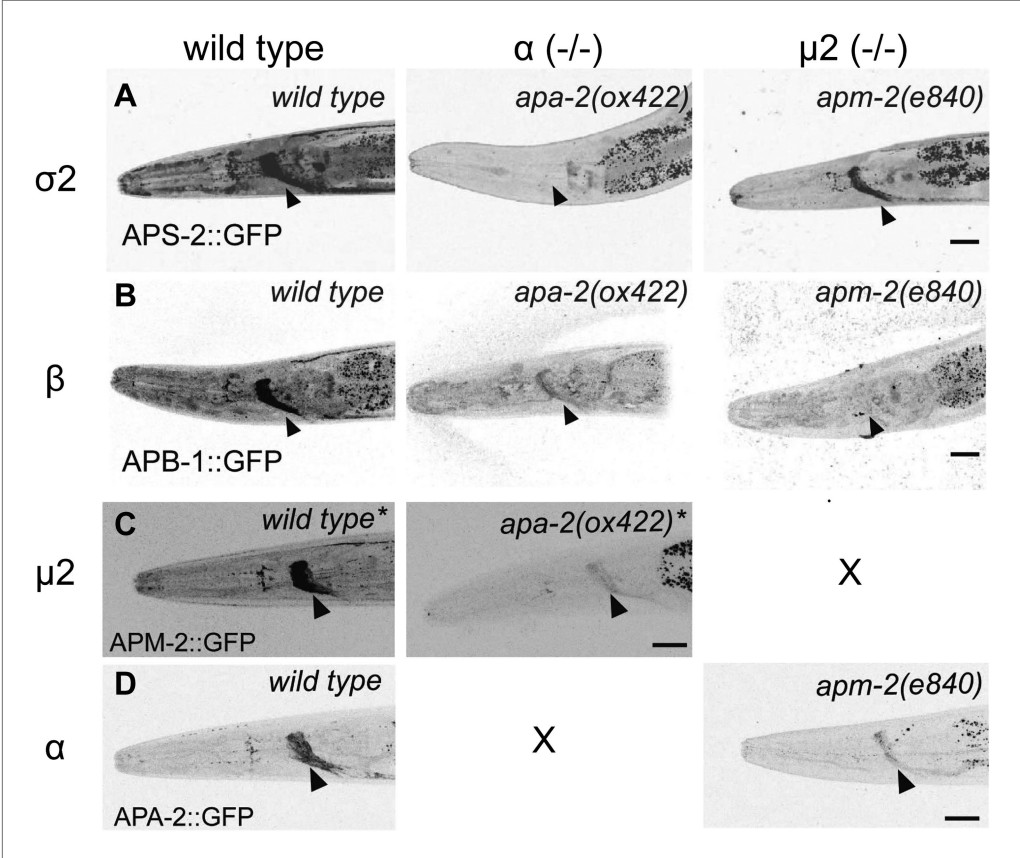

**Figure 6**. AP2 hemicomplexes are partially stable in vivo. All images are inverted to better visualize GFP fluorescence. (**A**) Synaptic localization of σ2 adaptin (APS-2::GFP) in α and μ mutants. The nerve ring is indicated by the black arrowhead. (**B**) Synaptic localization of β adaptin (APB-1::GFP) in α and μ2 mutants. (**C**) Synaptic localization of μ2 adaptin (APM-2::GFP) rescuing construct in μ2 mutants (labeled as *wild type\**) and an α μ2 double mutant (*apm-2(e840) apa-2(ox422) X*, labeled as *apa-2(ox422) \**). The single copy APM-2::GFP transgene *oxSi54* fully rescues the *apm-2(e840)* mutation. (**D**) Synaptic localization of α adaptin (APA-2::GFP) in *wild type* and an μ2 mutant *apm-2(e840)*. The scale bar represents 20 μm. Please refer to **Table 1** for detailed quantification.

The following figure supplements are available for figure 6:

**Figure supplement 1**. μ2-adaptin is present in α-adaptin mutants.

**Figure supplement 2**. σ2-adaptin is more unstable in α-adaptin than in μ2-adaptin mutants.

axons (**Nonet et al., 1999**). By contrast, in AP2 adaptin mutants, synaptic vesicle proteins are not grossly mislocalized. Synaptotagmin, the vesicular GABA transporter (UNC-47), and synaptogyrin are largely confined to synaptic varicosities in α-adaptin single mutants and α-μ2 adaptin double mutants, although the GFP signal is slightly diffuse in axons (**Figure 7A,B**). These data are consistent with previous data demonstrating that the relevant adaptors for synaptotagmin and the GABA transporter are Stonin and BAD-LAMP/UNC-46, respectively (**Schuske et al., 2007**; **Maritzen et al., 2010**; **Mullen et al., 2012**). These results suggest that vesicle proteins are endocytosed properly in AP2 mutants, although it is possible that some proteins remain on the surface but are confined to the synapse.

To determine if there is a defect in membrane endocytosis, we characterized the ultrastructure of neuromuscular junctions in α-adaptin mutants (**Figure 8A**). In mutants lacking *apa-2* in the nervous system, synaptic vesicle numbers are reduced to 71% in acetylcholine neurons and 59% in GABA neurons (**Figure 8B**; **Figure 8—figure supplement 1**). This moderate reduction in synaptic vesicle numbers is similar to the loss observed in μ2 mutants (**Gu et al., 2008**). The synaptic vesicle defects

**Table 1.** GFP fluorescence of tagged AP2 subunits in the nerve ring (quantification for **Figure 6**). Average GFP intensity in the nerve ring (percentage of the wild type)

|  | wild type | apa-2 | apm-2 |
|---|---|---|---|
| σ2::GFP | 3543 ± 169 (100%) | 389 ± 28 (11%) | 1391 ± 51 (39%) |
| β1::GFP | 3881 ± 31 (100%) | 2329 ± 123 (60%) | 1360 ± 62 (35%) |
| μ2::GFP | 2783 ± 142 (100%) | 532 ± 73 (19%) | – |
| α::GFP | 3230 ± 132 (100%) | – | 1296 ± 126 (40%) |

The data are mean ± SEM of averaged fluorescence, n = 5 worms each. The p value for all pair-wise comparisons (wild type vs mutants or *apa-2* vs *apm-2*) is p<0.0001. Student's *t* test. Note the beta subunit in *C. elegans* is shared by both the AP1 and AP2 complexes. Beta levels are reduced in apm-2 mutants compared to apa-2 mutants; however, beta is still present and stable in AP1 complexes in apm-2 mutants. In particular, beta is highly expressed in pharyngeal muscle, which is included in the region of interest.

observed in α-adaptin mutants can be fully rescued by expression of APA-2 in the nervous system. Defects in synaptic vesicle numbers are more severe in mutants lacking rescue in the skin (56% in acetylcholine neurons and 29% in GABA neurons compared to the wild type) as was observed in μ2 mutants (**Gu et al., 2008**). In summary, specific loss of just α-adaptin or just μ2-adaptin in neurons only leads to a moderate defect in synaptic vesicle number.

In contrast to the single mutants, complete loss of AP2 at synapses leads to a severe defect in synaptic vesicle number. In synapses of *apa-2(ox422) apm-2(e840)* double mutants (but rescued in the epidermis) the number of synaptic vesicles is reduced to 28% in acetylcholine neurons and 31% in GABA neurons (**Figure 8B**; **Figure 8—figure supplement 1**). It is likely that of loss of α and μ2 leads to a complete loss of AP2 since σ2 and β are lost at synapses in each of these mutants respectively (**Figure 6A,B**). In summary, loss of μ2 alone leads to a 31% decrease in synaptic vesicles in acetylcholine neurons (**Gu et al., 2008**), loss of α-adaptin alone leads to a 29% decrease, but a complete loss of AP2 leads to a 70% reduction in synaptic vesicle number. These data suggest that complete inactivation of AP2 requires removal of both the α and μ2 subunits.

The diameter of the remaining synaptic vesicles is slightly increased in *apa-2* and *apm-2 apa-2* double mutants (**Figure 8C**; **Figure 8—figure supplement 2**). The median diameter of synaptic vesicles in the wild type is 28.4 nm, the diameter in *apa-2* mutants (skin rescued) is 30.9 nm, and the diameter in *apa-2 apm-2* double mutants (skin rescued) is 31.9 nm (**Figure 8C**). These data suggest that the AP2 complex may play a role in regulating the size of synaptic vesicles. Alternatively, the effect on vesicle size may be indirect due to pleiotropic defects in endocytosis.

Beyond the slight increase in diameter of synaptic vesicles, α-adaptin mutant synapses also exhibit an accumulation of large vesicles (diameter > 40 nm, **Figure 8A,D,E**; **Figure 8—figure supplement 3**). This phenotype is not apparent in μ2 mutants (**Gu et al., 2008**), suggesting different roles for α and μ2 at synapses. In *apa-2 apm-2* double mutants a large vesicle is often observed adjacent to the dense projection and very large vesicles occupy the center of the synaptic varicosity (**Figure 8A,E, F**; **Figure 8—figure supplement 4**). We speculate that these large vesicles could be endosomal intermediates generated by bulk endocytosis.

## Exocytosis is proportional to synaptic vesicle number in AP2 mutants

Are these large vesicles *bonafide* synaptic vesicles? Specifically, can they fuse and release neurotransmitter in an electrophysiological assay? The increase in diameter of synaptic vesicles was accompanied by an increase in the amount of neurotransmitter released by a synaptic vesicle. Miniature postsynaptic currents ('minis') were measured from motor neurons using voltage-clamp recordings from body muscles (**Figure 9A,B**). In *apa-2(ox422)* mutants, the amplitude from miniature spontaneously released vesicles (minis) is increased by 40% (**Figure 9C**). The mini amplitudes in the skin-rescued single and double mutants are also larger, although they do not reach statistical significance. The enhanced mini amplitude could have been caused by an increase in postsynaptic receptor density due to a defect in receptor endocytosis (**Kittler et al., 2005**; **Kastning et al., 2007**; **Vithlani and Moss, 2009**). However, the defect in mini amplitude was fully rescued by expressing *apa-2* in neurons (**Figure 9C**). Thus, the increase in mini current amplitude is consistent with the observed increase in the diameter of synaptic vesicles.

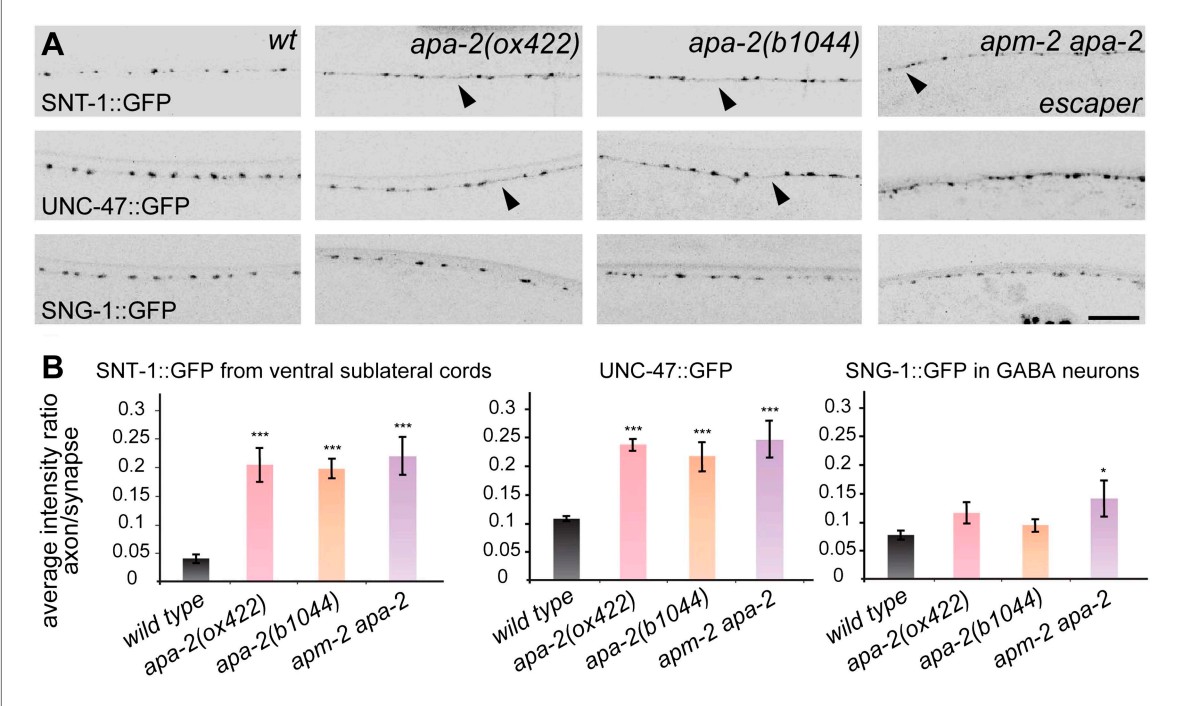

**Figure 7**. α-adaptin mutants exhibit weak defects in synaptic vesicle protein localization. (**A**) All images are inverted to better visualize GFP fluorescence. Synaptic localization of synaptic vesicle proteins in *apa-2* and *apm-2(e840) apa-2(ox422)* double mutants (an escaper with no skin rescue). Synaptotagmin (SNT-1::GFP) is expressed in all neurons under its own promoter and imaged in ventral sublateral cords. VGAT (UNC-47::GFP) and synaptogyrin (SNG-1::GFP) are expressed in GABA neurons and imaged in the dorsal nerve cord. Presynaptic varicosities of neuromuscular junctions along the nerve cords of an adult hermaphrodite are visible as fluorescent puncta. The axon regions with increased fluorescence are indicated by black arrowheads. Images are confocal Z-stack projections through the worm nerve cord. The scale bar represents 10 μm. (**B**) Quantification of the average fluorescence intensity ratio between axon region and synaptic region. Ratio of SNT-1::GFP mean ± SEM: wild type 0.041 ± 0.007 n = 10, *apa-2(ox422)* 0.204 ± 0.029 n = 10 (p<0.0001), *apa-2(b1044)* 0.196 ± 0.017 n = 10 (p<0.0001), *apa-2(ox422) apm-2(e840)* 0.219 ± 0.032 n = 6 (p<0.0001). Ratio of UNC-47::GFP mean ± SEM: wild type 0.108 ± 0.004 n = 8, *apa-2(ox422)* 0.239 ± 0.009 n = 8 (p<0.0001), *apa-2(b1044)* 0.220 ± 0.024 n = 7 (p=0.0003), *apa-2(ox422) apm-2(e840)* 0.247 ± 0.032 n = 6 (p=0.0003). Ratio of SNG-1::GFP mean ± SEM: wild type 0.077 ± 0.008 n = 8, *apa-2(ox422)* 0.116 ± 0.019 n = 10 (p=0.1026), *apa-2(b1044)* 0.095 ± 0.011 n = 10 (p=0.2252), *apa-2(ox422) apm-2(e840)* 0.143 ± 0.032 n = 5 (p=0.0308). * p<0.05, ** p<0.01, *** p<0.001.

The following figure supplements are available for figure 7:

**Figure supplement 1**. α-adaptin colocalizes with synaptobrevin at synapses.

The reduction in synaptic vesicle numbers was also paralleled by a reduction in the electrophysiological response of the neuromuscular junctions. There is a 25% reduction in the amplitude of evoked release in skin-rescued *apa-2* mutants, and this defect can be rescued by expressing *apa-2* in neurons. The double mutants exhibit a more severe, 42% reduction in the amplitude of the evoked responses (***Figure 9E***). There is also a more severe reduction in the rates of tonic synaptic vesicle fusion. Skin-rescued *apa-2* animals exhibit a 50% reduction in mini frequency, and the skin-rescued *apa-2 apm-2* double mutants exhibit a 68% reduction in mini frequency (***Figure 9D***). The reduction in vesicle fusions (68% reduction) is proportional to the reduction in synaptic vesicle numbers at synapses (70% reduction), suggesting that the vesicles seen by electron microscopy are *bonafide* synaptic vesicles in the AP2 mutants. In summary, the loss of both α- and μ2-adaptin leads to a more severe synaptic defect than the single mutants, suggesting that these subunits can function independently in synaptic vesicle endocytosis.

## Discussion

In this study, we genetically characterized AP2 function in *C. elegans* with a particular focus on the synapse. The results indicate that AP2 can function as two hemicomplexes comprised of either the α/σ subunits or μ/β subunits. The evidence for hemicomplexes is the following: First, in α-adaptin

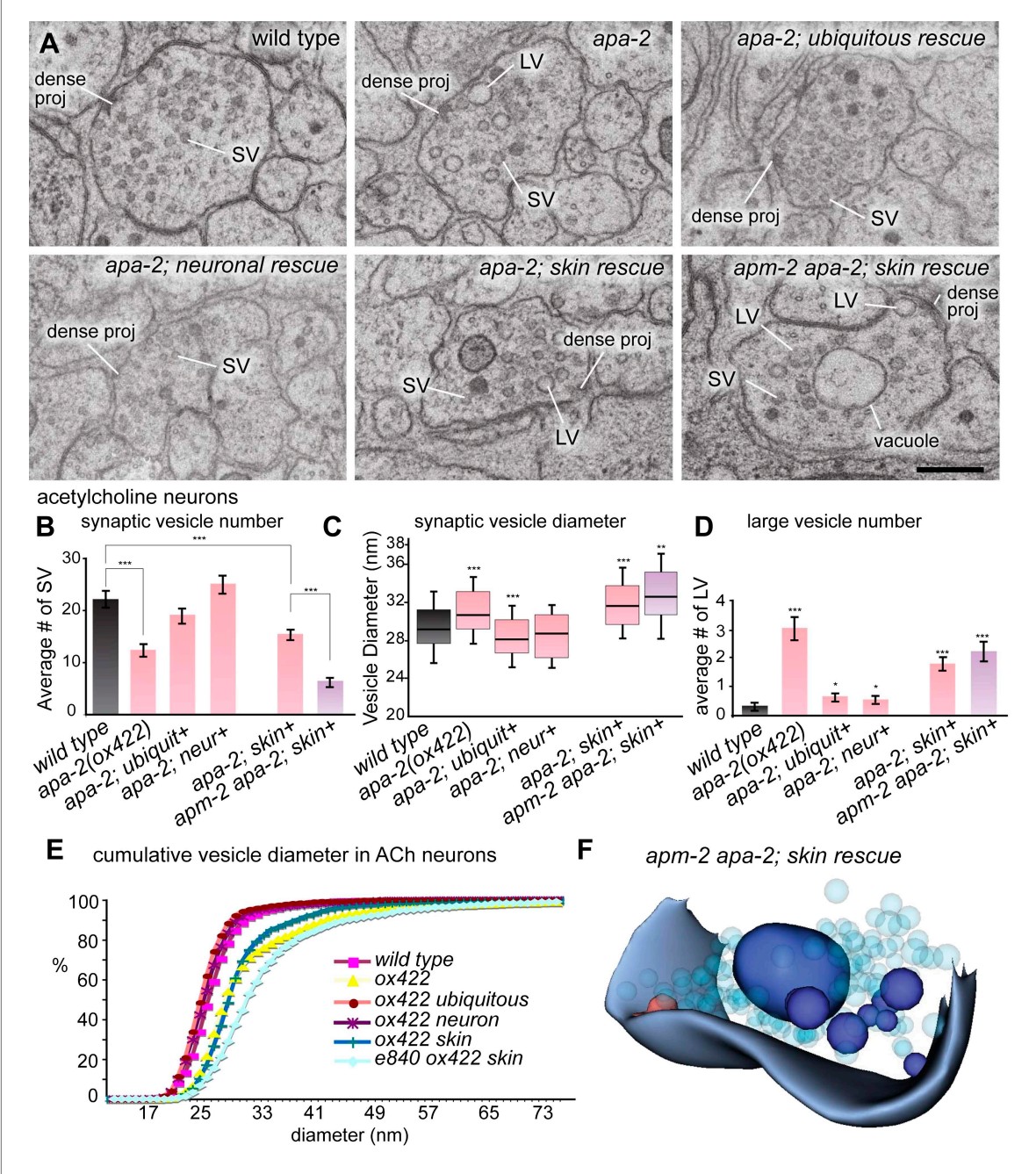

**Figure 8**. Large vesicles accumulate at synapses in AP2 mutants. (**A**) Representative images of acetylcholine neuromuscular junctions in the ventral nerve cord from the wild type, *apa-2(ox422)*, ubiquitously-rescued *apa-2(ox422)*, neuronally-rescued *apa-2(ox422)*, skin-rescued *apa-2(ox422)*, skin-rescued *apa-2(ox422) apm-2(e840)* in adult hermaphrodites. At *apa-2 apm-2* synapses, at least one large vesicle was usually observed adjacent to the dense projection (13/21 synapses), and a large vacuole in the center of the varicosity (17/21 synapses). The scale bar represents 200 nm. Abbreviations: SV: synaptic vesicle; LV: large vesicle; dense proj: dense projection. (**B**) Morphometry of acetylcholine neuromuscular junctions in adaptin mutants. The number of synaptic vesicles is reduced in neurons lacking α adaptin or both α and μ2-adaptins. Average number of synaptic vesicles per profile containing a dense projection ± SEM n = synapses: wild type 22.0 ± 1.4 n = 35, *apa-2(ox422)* 12.3 ± 1.1 n = 66 (p<0.0001), ubiquitous rescued *apa-2(ox422)* 19.1 ± 1.1 n = 54 (p=0.1052), neuron-rescued *apa-2(ox422)* 25.1 ± 1.5 n = 49 (p=0.1501), skin-rescued *apa-2(ox422)* 15.6 ± 0.8 n = 97 (p<0.0001), skin-rescued *apa-2(ox422) apm-2(e840)* 6.2 ± 0.8 n = 47(p<0.0001; compared with skin rescued *apa-2(ox422)* p<0.0001). (**C**) Median size of synaptic vesicles per profile containing a dense projection n = synapses: wild type 28.4 nm n = 9, *apa-2(ox422)* 29.9 nm n = 12 (p=0.0007), ubiquitous rescued *apa-2(ox422)* 27.4 nm n = 12 (p=0.0001), neuron- rescued *apa-2(ox422)* 27.9 nm n = 12 (p=0.5538), skin-rescued *apa-2(ox422)* 30.9 nm n = 23 (p<0.0001), skin-rescued *apa-2(ox422) apm-2(e840)* 31.9 nm n = 11 (0.0057). Median is the middle line and box defines the 25th and 75th percentiles. The length of the whiskers indicates the span between the 10th and 90th percentiles. (**D**) Average number of large vesicles (clear core and the diameter > 35 nm) per

*Figure 8. Continued on next page*

*Figure 8. Continued*

profile containing a dense projection ± SEM n = synapses: wild type 0.30 ± 0.11 n = 30, *apa-2(ox422)* 3.1 ± 0.34 n = 72 (p<0.0001), ubiquitous rescued *apa-2(ox422)* 0.7 ± 0.13 n = 43 (p=0.0302), neuron-rescued *apa-2(ox422)* 0.6 ± 0.08 n = 62 (p=0.0325), skin-rescued *apa-2(ox422)* 1.8 ± 0.2 n = 97 (p<0.0001), skin-rescued *apa-2(ox422) apm-2(e840)* 2.2 ± 0.29 n = 53 (p<0.0001). (**E**) Cumulative vesicle diameter in acetylcholine neurons. For all panels, the imaged synapses are from two young adult hermaphrodites for each genotype. Statistics are comparison with wild type, except where marked. * p<0.05, ** p<0.01, *** p<0.001. (**F**) 3D modeling of an acetylcholine synapse from a skin-rescued *apa-2(ox422) apm-2(e840)* animal. Structures were hand-traced from ten consecutive sections using an imageJ plugin, TrakEM2 (**Cardona et al., 2012**). The transparent light-blue structures are synaptic vesicles, and the red structure is a dense projection. Large vesicles (dark blue) that accumulate in the terminal are typically severed from the surface.

The following figure supplements are available for figure 8:

**Figure supplement 1**. Synaptic vesicles are reduced at GABA synapses in α-adaptin mutants and α-adaptin μ2-adaptin double mutants.

**Figure supplement 2**. Synaptic vesicle diameters are larger in α-adaptin mutants.

**Figure supplement 3**. Large vesicles accumulate in α-adaptin mutants and α-adaptin μ2-adaptin double mutants.

**Figure supplement 4**. In some cases, the large vacuole remains associated with the plasma membrane in α-adaptin μ2-adaptin double mutants.

mutants, the μ2-β subunits are stable, but the small σ2 subunit is unstable. Second, in μ2-adaptin mutants, α-σ2 subunits are stable, but the β subunits are unstable. Third, specific cargoes require the cognate hemicomplex for endocytosis. Fourth, the subunits contribute genetically independent functions to viability, body morphology and synaptic vesicle biogenesis. Although our data suggest that AP2 hemicomplexes can function in *C. elegans*, it must be emphasized that they are not fully independent; each hemicomplex is less stable in the absence of the other. Nonetheless, a complete block of AP2 function requires the simultaneous removal of both α- and μ2-adaptins.

Below, we discuss four aspects of these results: What is the structural basis for hemicomplex function? What is the functional division of hemicomplexes? Can hemicomplexes function in other organisms? How can synapses function in the absence any AP2 function?

Recent structural studies support the possibility of stable hemicomplexes. Previous trypsin-sensitivity experiments suggested that AP2 undergoes a conformational change between the cytosolic and clathrin-bound states (**Matsui and Kirchhausen, 1990**). The crystal structures of both the closed and open conformations have been solved (**Collins et al., 2002**; **Jackson et al., 2010**). In the open state, μ2-adaptin is postulated to undergo a large-scale conformational change; this rearrangement brings the four PIP2 binding sites and two endocytic motif binding sites of AP2 into a single plane. In this open conformation, the interactions between the C-terminal μ2 domain and α and σ2 are lost, and the binding surface between the C-terminal domain of μ2 and β is doubled (**Jackson et al., 2010**). This implies that upon cargo binding, the interaction between μ2 and β is strengthened while the contacts with the other half of the complex are weakened. It is possible that upon cargo binding the AP2 complex becomes two loosely connected hemi-complexes.

What is the functional relationship between the hemicomplexes? There are three possibilities: inseparable functions, separable functions, and redundant functions. First, some functions seem to require both hemi-complexes combined, and it is surprising that loss of one hemicomplex does not eliminate all AP2 function. For example, recruitment of AP2 to membranes in vitro requires both PIP2 binding sites on α and β subunits (**Jackson et al., 2010**). On the other hand, PIP2 binding sites on each of the hemicomplexes may be sufficient for membrane association albeit with a lowered avidity. Second, other AP2 functions may be uniquely provided by each hemicomplex. Substrate binding in some cases is subunit-specific and loss of one hemicomplex preferentially affects a cargo protein, for example, MIG-14 is not recruited in a μ2 adaptin mutant. On the cytoplasmic side, the ear of the alpha subunit preferentially binds particular ancillary proteins like amphiphysin, synaptojanin, Numb, and stonin2 (**Owen et al., 2000**; **Santolini et al., 2000**; **Praefcke et al., 2004**; **Jung et al., 2007**), whereas clathrin heavy chain binds the appendage of β strongly and only binds the α appendage weakly (**Shih et al., 1995**; **Owen et al., 2000**; **Schmid et al., 2006**). Thus, loss of a single hemicomplex will result in the loss of only a specific subset of AP2 functions. Third, some functions might be mediated by either subunit, and only a double mutant would lead to a severe phenotype. For example, the

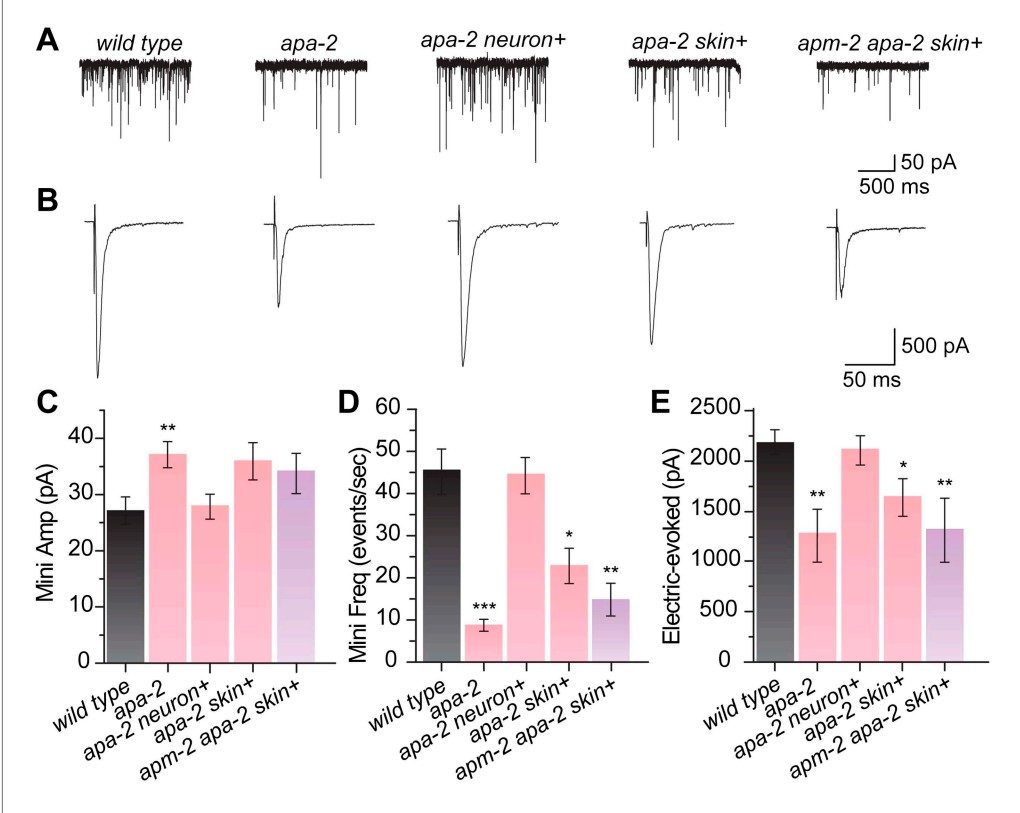

**Figure 9**. Synaptic vesicle fusion is reduced in α adaptin mutants. (**A**) Sample traces of miniature postsynaptic current (minis) recorded from the wild type, *apa- 2(ox422)*, *apa-2(ox422)* neuronal-rescued, *apa-2(ox422)* skin-rescued and *apa-2(ox422) apm-2(e840)* skin-rescued worms. (**B**) Sample traces of evoked postsynaptic current (electrically evoked) recorded from same genotypes. (**C**) Summary of mini amplitudes (pA ± SEM n = animals): wild type 26.4 ± 2.5 n = 16, *apa-2(ox422)* 36.9 ± 2.5 n = 19 (p=0.0058), *apa-2(ox422)* skin-res. 35.2 ± 3.6 n = 9 (p=0.0516), *apa-2(ox422)* neur-res. 28.1 ± 2.4 n = 21 (p=0.6313), *apa-2(ox422) apm-2(e840)* skin-res. 33.3 ± 3.8 n = 9 (p=0.1287). (**D**) Summary of mini frequency (minis/sec ± SEM n = animals): wild type 43.9 ± 5.9 n = 16, *apa-2(ox422)* 7.8 ± 1.5 n = 19 (p<0.0001), *apa-2(ox422)* skin-res. 22.5 ± 4.5 n = 9 (p=0.0206), *apa-2(ox422)* neur-res. 44.9 ± 4.8 n = 21(p=0.8951), *apa-2(ox422) apm-2(e840)* skin-res. 14.2 ± 4.0 n = 9 (p=0.0019). (**E**) Summary of evoked amplitude (pA ± SEM n = animals): wild type 2159.6 ± 131.1 n = 11, *apa-2(ox422)* 1259.1 ± 274.9 n = 5 (p=0.0044), *apa-2(ox422)* skin-res. 1627.3 ± 182.0. n = 6 (p=0.0303), *apa-2(ox422)* neur-res. 2090.7 ± 149.0 n = 6 (p=0.7468), *apa-2(ox422) apm-2(e840)* skin-res. 1264.3 ± 323.7 n = 6 (p=0.0082). * p<0.05, ** p<0.01, *** p<0.001.

appendage domains of both large subunits bind some of the same proteins, for example AP180, epsin, and eps15 (***Owen et al., 2000***; ***Mishra et al., 2004***). Importantly, redundancy need not act only at the level of AP2 subunits but could be contained within the network of associated proteins. Although clathrin is largely recruited to AP2 by the β subunit, even in the absence of β, it could still be recruited indirectly to the complex via AP180—the web of interactions within the clathrin complex generates a redundant network (***Royle, 2006***; ***Schmid et al., 2006***). Thus, in contrast to un-networked hubs (***Jeong et al., 2001***), the loss of the hub does not cause things to fall apart; the center can hold.

Are functional hemicomplexes conserved? Certainly the sequences of the AP2 subunits are strongly conserved. For example, the amino acid sequences of AP2 subunits in *C. elegans* and mouse are at least 64% identical (α 65%; β2 64%; μ2 82%; σ2 95%). It is also possible that the ability of AP2 hemicomplexes to function is also conserved. A purified human α-σ2 hemicomplex can bind di-leucine motifs, suggesting that hemicomplexes can be stable and exhibit appropriate biochemical interactions (***Doray et al., 2007***). Although double mutants have not been analyzed in other metazoans, a genome-wide genetic interaction analysis in *S. pombe* found that mutations in AP2 β2 and σ2 exhibited synthetic interactions in double mutants (***Frost et al., 2012***), suggesting that functional hemicomplexes may be conserved in other organisms. On the other hand, knocking down the μ2 subunit in cultured

hippocampal neurons caused a concomitant 96% reduction of α-adaptin suggesting that hemicomplexins are not stable in these cells (*Kim and Ryan 2009*). It is likely that the stability of hemicomplexes may vary in organisms depending on a variety of factors such as temperature, chaperones and degradation machinery.

What is the molecular role of AP2 in synaptic vesicle biogenesis? A reduction of synaptic vesicle numbers by 70% and the accumulation of large vesicles imply an important role of AP2 in endocytosis. These defects resemble those observed in synaptotagmin mutants in *C. elegans* or after acute disruption of synaptotagmin in *Drosophila* (*Jorgensen et al., 1995*; *Poskanzer et al., 2006*). Moreover, the synaptic phenotypes of mutants lacking stonin are similar (*Fergestad et al., 1999*; *Mullen et al., 2012*). One possibility is that AP2 nucleates synaptic vesicle endocytosis with stonin and synaptotagmin. The synaptotagmin C2B domain binds AP2 via the mu-homology domain of μ2-adaptin (*Zhang et al., 1994*; *Haucke et al., 2000*), and the C2A domain binds the mu-homology domain of stonin (*Jung et al., 2007*). It is possible that these proteins work together in a single process. In fact analysis of double mutants suggest that stonin and AP2 act in a similar process (*Mullen et al., 2012*). In the simplest model, synaptotagmin recruits stonin and AP2 to the plasma membrane to recover synaptic vesicle components.

On the other hand, the AP2 double mutants lacking both α- and μ2-adaptins exhibit remarkably normal locomotion and evoked currents. One is forced to conclude that despite an important role in endocytosis, that synaptic vesicles are still being generated in the absence of AP2. What process contributes to synaptic vesicle endocytosis when AP2 is missing? One possibility is that AP1 or AP3 could compensate for the loss of AP2. In the mouse, there is evidence that AP1 could function at the synapse and substitute for AP2 in its absence (*Kim and Ryan, 2009*; *Glyvuk et al., 2010*). Alternatively AP3 might be able to provide function in the absence of AP2 (*Blumstein et al., 2001*; *Voglmaier et al., 2006*). However, AP1 and AP3 are not likely to be recycling vesicles from the membrane at the *C. elegans* neuromuscular junctions. First, the presence of β-adaptin (shared by AP1) in the nerve ring is completely dependent upon the presence of μ2; AP1 does not seem to be at the synapse (*Figure 6B*). Second μ2-μ3 double mutants do not exhibit a synthetic phenotype, suggesting that AP3 does not substitute for AP2 at *C. elegans* synapses (*Gu et al., 2008*).

It is more likely that the adaptor protein associated with the AP2 complex, such as AP180, mediates endocytosis in the absence of the AP2 complex. AP180 possesses functions remarkably similar to AP2. AP180 can bind and stimulate clathrin assembly and bind PIP2 in the membrane (*Hao et al., 1999*; *Ford et al., 2001*). It acts as an adaptor for synaptic vesicle proteins since it can bind and recruit synaptobrevin to invaginating vesicles (*Nonet et al., 1999*; *Miller et al., 2011*). Finally AP180 mutants in *C. elegans* exhibit defects in synaptic vesicle endocytosis as analyzed by electron microscopy (*Nonet et al., 1999*). It is possible that the remaining functional synaptic vesicles in the absence of AP2 are generated by AP180.

Where then does AP2 act? Classic studies of synaptic ultrastructure of frog and fly synapses suggest that clathrin and the AP2 complex act at the plasma membrane (*Heuser and Reese, 1973*; *Gonzalez-Gaitan and Jackle, 1997*). The data presented here do not contradict those studies, but also suggest that synaptic vesicle proteins and membrane can be recovered from the membrane despite a loss of AP2. The most prominent defect observed in AP2 mutants is the presence of large diameter vesicles and vacuoles in the synapses of the α-μ2 adaptin double mutants. In some cases these vacuoles appear to be attached to the plasma membrane at the adherens junctions (*Figure 8—figure supplement 4*), suggesting a defect in the formation and cleavage of vesicles from the plasma membrane. In other cases, reconstructions of these vacuoles from serial electron micrographs indicate that they are separated from the plasma membrane (*Figure 8F*). These data suggest that AP2 has a late function; AP2 may be required to regenerate synaptic vesicles from endosomes.

In summary, these data suggest two major conclusions: First, the AP2 complex can function as two semi-independent hemicomplexes, consistent with new structural data for the complex. Second, there are at least two mechanisms (AP2-dependent and AP2-independent) for endocytosis at synapses in *C. elegans* that regenerate synaptic vesicles and maintain synaptic function.

## Materials and methods

### Strains and screens

The wild type is Bristol N2. All other genotypes are described in *supplementary file 1*. *apa-2(b1044)* was isolated by polymerase chain reaction and sibling selection from an ultraviolet- and trimethyl-psoralen-mutagenized 'mutant library' generously provided by X Li, A Melendez and I Greenwald. Primers

ATTTGTCGGTCGGTACTTGC and ATTCGCCTACGCCATTCTTC were used in the first round of amplification, whereas the nested primers ATCTGTCGTAATTGTCACGG and TTTGGATCCACGTCAGTCAG were used for the second round of amplification. The reference strain EG4739 for *apa-2(b1044) X* was outcrossed twice before phenotypic analysis. *apa-2(ox422)* was isolated from a non-complementation screen of *b1044* from 4000 haploid genomes mutagenized by ENU. *ox422* is an A to T transversion that creates a premature stop at lysine 215. A second deletion allele *ox421* was also isolated which removes the entire *apa-2* ORF. *ox421* is likely to be a deficiency since it removes at least 6 kb upstream and 3 kb downstream of the *apa-2* ORF and thus deletes genes upstream and downstream of *apa-2*. The reference strain EG6147 for *apa-2(ox422) X* was outcrossed seven times before phenotypic analysis. All *oxSi* single copy insertions were generated by MosSCI (*Frokjaer-Jensen et al., 2008*).

## GFP and MosSCI constructs

*apa-2* translational reporter: 1.9 kb promoter and the coding sequence of *apa-2* was cloned into pGEM-3Zf vector. This fragment was fused to GFP-*unc-54* 3'UTR at the XbaI and HindIII sites.

Three-fragment Multisite Gateway vectors were used (Invitrogen, Grand Island, NY; catalog no.12537-023) for generating most other constructs. PENTRY4-1 was used as the slot 1 promoter entry vector. Promoters include P*dpy-30*, P*dpy-7*, P*rab-3* and P*unc-47*; the promoter fragments do not include the initiating methionine codon (ATG). PENTRY1-2 was used as the slot 2 ORF entry vector, ORFs include *apa-2(cDNA)*, *apm-2(cDNA)*, *apb-1(cDNA)*, *eGFP::CD4(di-leucine)*, and *sng-1(cDNA)*, all of which have an ATG immediately following the *att* site at the beginning of the ORF; they do not have a stop codon at the 3' end. PENTRY2-3 was used for the slot 3 C-terminal tag and 3'UTR entry vector, slot 3 clones include *GFP-unc-54 3'UTR* and *mCherry-unc-54 3'UTR.* The destination vectors are Gateway pDEST R4-R3, pCFJ150 for MosSCI on chromosome II and pCFJ201 for MosSCI on chromosome IV (*Frokjaer-Jensen et al., 2008*).

The 1.2 kb *aps-2* promoter and the 1 kb *aps-2* genomic coding sequence were cloned by PCR from wild-type genomic DNA. GFP with the *unc-54* 3'UTR was fused to the C-terminus of *aps-2* using a PstI site and the entire fusion fragment was dropped between the restriction sites BssHII and SpeI on pCFJ151 for MosSCI on chromosome II.

## Microinjection

The final DNA concentration of each injection mix was 100 ng/µl. This target concentration was obtained with the addition of Fermentas 1 kb DNA ladder (#SM0311).

APA-2 translational GFP: pMG16 *apa-2::GFP* was injected into *lin-15(n765ts) X* animals at 1 ng/µl. The coinjection marker *lin-15(+)* was used at 50 ng/µl.

APA-2 and synaptobrevin colocalization: pRH324 P*unc-47::SNB-1::tagRFP* was injected into the wild type (N2) at 0.25 ng/µl. The co-injection marker P*unc-122::GFP* was used at 50 ng/µl. F1 transgenic worms were singled. One of the transgenic lines, *oxEx1411,* was crossed into *dkIs160[Punc-25::GFP::APA-2; unc-119(+)]*.

*apa-2 apm-2* double mutant skin rescue: pMG50 P*dpy-7::APM-2::GFP* and pMG40 P*dpy-7::APA-2::mCherry* were coinjected into the adaptin double mutant-balanced strain EG6158 *+/szT1[lon-2(e678)] I; szT1/apm-2(e840) apa-2(ox422) X* at 1 ng/µl each. The coinjection marker was P*unc-122::GFP,* at 50 ng/µl. In the next generation, rescued but egg-laying defective worms were singled. One of the lines, EG6151, was used in the electron microscopy and electrophysiology assays.

## Western blot analysis

Worm samples were prepared by boiling 1 volume of worm pellet in 1 volume of 2× loading buffer for 5 min. Samples were run on a 10% SDS-PAGE gel and then transferred to PVDF transfer membrane (Immobilon). The primary antibody for adaptin was a rabbit polyclonal anti-APA-2 (*Sato et al., 2005*) at a dilution of 1:500. Primary antibody incubation was done in 5% BSA at 4°C overnight. The primary antibody for the anti-tubulin control was 12G10 mouse monoclonal anti-tubulin (Developmental Studies Hybridoma Bank) at a dilution of 1:10,000. Primary antibody incubation was done in 5% BSA at room temperature for 1 hr. Secondary antibodies were anti-rabbit and mouse IgG fragments conjugated with HRP (GE Healthcare, Pittsburgh, PA). Secondary incubations were done in 5% BSA at room temperature for 45 min. The detection reagent was SuperSignal West Dura (Thermo Scientific, Waltham, MA).

For anti-GFP western blot, the primary antibody for GFP was mouse monoclonal anti-GFP at a dilution of 1:5000 (Clontech, Mountain View, CA; Cat. No. 632375). Primary antibody incubation was done in 5% sea block blocking buffer (Pierce, Rockford, IL; prod#37,527) at 4°C overnight.

## Fertilized embryo quantification

For each genotype, 10–12 L4 worms were singled to plates and were transferred to a fresh plate every 12 hr. The transfers stopped when the worm burst (due to an egg-laying defect such as in AP2 mutants) or the worm started laying unfertilized oocytes (such as wild type). The fertilized embryos from each animal were counted to determine the brood size. If the worm was lost during the transfer, the data were discarded. *apm-2 apa-2* double mutants were survivors from the balanced strain EG6158 *+/szT1[lon-2(e678)] I; szT1/apm-2(e840) apa-2(ox422) X* .

## Embryonic lethality

All embryos from the brood size quantification were scored for hatching. Hatching was checked after 12 hr. Unhatched embryos were marked and checked again after another 12 hr. The total dead embryos were divided by the brood size to determine the lethal fraction.

## Developmental time quantification

L1 worms were picked to a plate and checked every 12 hr for the growth until they reached L4 stage. If the L4 stage was difficult to score due to the sickness of the worm, the scoring was confirmed 12 hr later to insure the animal had become an adult.

## Confocal microscopy

Worms were immobilized using 2% phenoxypropanol and imaged on a Zeiss Pascal LSM5 confocal microscope using a plan-Neofluar 10× 0.3 NA, 20× 0.5 NA, 40× 1.3 NA oil or Zeiss plan-apochromat 63× 1.4NA oil objectives.

## Electron microscopy

Adult nematodes were prepared in parallel for transmission electron microscopy as previously described (*Hammarlund et al., 2007*). To briefly summarize, 10 young adult hermaphrodites were placed into a freezing cup (100 μm well of type A specimen carrier) containing space-filling bacteria, covered with a type B specimen carrier flat side down, and frozen instantaneously in the BAL-TEC HPM 010 (BAL-TEC, Liechtenstein). The frozen animals were fixed in the Leica AFS device with 1% osmium tetroxide and 0.1% uranyl acetate in anhydrous acetone for 2 days at −90°C and for 38.9 more hr with a gradual increase in temperature (5 °C/hr to −20°C over 14 hr, constant temperature at −20°C for 16 hr, and 10 °C/hr to 20°C over 4 hr). The fixed animals were embedded in epon-araldite resin following the infiltration series (30% epon-araldite/acetone for 4 hr, 70% epon-araldite/acetone for 5 hr, 90% epon-araldite/acetone overnight, and pure epon-araldite for 8 hr). Mutant and control blocks were blinded. Ribbons of ultra-thin (33 nm) serial sections were collected using an Ultracut six microtome at the level of the anterior reflex of the gonad. Images were obtained on a Hitachi H-7100 electron microscope using a Gatan digital camera. Two hundred and fifty ultra-thin, contiguous sections were cut, and the ventral nerve cord was reconstructed from two animals representing each genotype. Image analysis was performed using Image J software. The numbers of synaptic vesicles (~30 nm), dense-core vesicles (~40 nm) and large vesicles (>40 nm) in each synapse were counted. Their distances from the presynaptic specialization and the plasma membrane, as well as their diameters, were measured in acetylcholine neurons VA and VB and the GABA neuron VD. A synapse 'profile' is defined as a single section that passes through the dense projection at a neuromuscular junction. Profiles are used for quantifying synaptic vesicle numbers at synapses (in fact, it uses a section that passes through the middle of the synapse as a representative section of the synapse). A 'synapse' encompasses adjacent serial sections containing a dense projection (usually four sections). Sections on either side of that density were also included if they contained synaptic vesicle numbers above the average number of synaptic vesicles per profile. 'Synapse' reconstructions are used for quantifying the presence of large vesicles or vacuoles associated with the dense projection; since there is usually only one such structure per synapse, partial reconstructions of the synapse are required to reliably identify these structures. Two-tailed Student's *t*-test was used for vesicle numbers and Mann-Whitney U test was used for vesicle diameters.

For synaptic modeling, we aligned 10 consecutive sections of an acetylcholine neuron from *apm-2 apa-2* double mutant using an imageJ plugin called TrakEM2 (*Cardona et al., 2012*, http://www.ncbi. nlm.nih.gov/pubmed/22723842). Plasma membranes, large vacuole membranes, and dense projections were traced using a paint brush tool. Synaptic vesicles and large vesicles are created using a 'ball' tool. The size of each vesicle is set by its diameter. The reconstructed volume was displayed in the

3D viewer. The plasma membrane and vacuole membrane were smoothed multiple times. The transparency of synaptic vesicles was set to 20%.

## Electrophysiology

*C. elegans* were grown at room temperature (22–24 °C) on agar plates with a layer of OP50 *Escherichia coli*. Adult hermaphrodite animals were used for electrophysiological analysis. Postsynaptic currents (mPSCs and ePSCs) at the NMJ were recorded as previously described (*Richmond et al., 1999*; *Liu et al., 2007*). To recapitulate, an animal was immobilized on a sylgard-coated glass coverslip by applying a cyanoacrylate adhesive along the dorsal side. A longitudinal incision was made in the dorsolateral region. After clearing the viscera, the cuticle flap was folded back and glued to the coverslip, exposing the ventral nerve cord and two adjacent muscle quadrants. A Zeiss Axioskop microscope equipped with a 40× water immersion lens and 15× eyepieces were used for viewing the preparation. Borosilicate glass pipettes with a tip resistance of 3–5 MΩ were used as electrodes for voltage clamping. The classical whole-cell configuration was obtained by rupturing the patch membrane of a gigaohm seal formed between the recording electrode and a body wall muscle cell. The cell was voltage-clamped at –60 mV to record mPSCs and ePSCs. ePSCs were evoked by applying a 0.5 ms square wave current pulse at a supramaximal voltage (25 V) through a stimulation electrode placed in close apposition to the ventral nerve cord. Postsynaptic currents were amplified with a Heka EP10 amplifier (InstruTECH) and acquired with Patchmaster software (HEKA). Data were sampled at a rate of 10 kHz after filtering at 2 kHz. The recording pipette solution contained the following (in mM): 120 KCl, 20 KOH, 5 TES, 0.25 $CaCl_2$, 4 $MgCl_2$, 36 sucrose, 5 EGTA, and 4 Na2ATP. The pH was adjusted to 7.2 with KOH, and the osmolarity was 310–320 mOsm. The standard external solution included the following (in mM): 150 NaCl, 5 KCl, 5 $CaCl_2$, 1 $MgCl_2$, 5 sucrose, 10 glucose and 15 HEPES, with the pH adjusted to 7.35 using NaOH and an osmolarity of 330–340 mOsm.

The amplitude and frequency of mPSCs were analyzed using MiniAnalysis (Synaptosoft, Decatur, GA). A detection threshold of 10 pA was used in initial automatic analysis, followed by visual inspections to include missed events (≥5 pA) and to exclude false events resulting from baseline fluctuations. Amplitudes of ePSCs were measured with Fitmaster (HEKA). The amplitude of the largest peak of ePSCs from each experiment was used for statistical analysis. Data were imported into Origin, version 7.5 (OriginLab, Northampton, MA), for graphing and statistical analysis. An unpaired t test was used for statistical comparisons. A value of $p < 0.05$ is considered statistically significant. All values are expressed as the mean ± the SEM n is the number of worms from which recordings were taken.

## Thrashing assay

A single worm was placed into a 50 µl drop of M9 solution. The worm was allowed to adapt to the liquid environment for 2 min. The number of body bends was counted for 60 s for each genotype (n = 5). A single body bend is considered a complete left to right and back to left bend. Two-tailed Student's *t* test was used for the statistics.

## Dumpy phenotype

Ten L4 stage worms were imaged for each genotype. The built-in measure function of LSM image browser (Zeiss) was used for the body-length quantification. Two-tailed Student's *t* test was used for the statistics.

## Image inversion and quantification

For fluorescence images, the figure panels were assembled as a single image then inverted and contrast adjusted evenly for better visualization. All individual images within the panel were treated identically.

All nerve ring images were exported as 12-bit RGB files. ImageJ 1.43u was used for quantification. The region of interest of fixed size was placed over the center of the nerve ring and fluorescence quantified. A region outside of the worm was used to quantify background fluorescence and the value was subtracted from the fluorescence image.

All worm nerve cord images were exported as 8-bit RGB files and ImageJ 1.43u was used for quantification. The region of interest was drawn by hand. The total pixel intensity and the total number of pixels were recorded to calculate the average fluorescence intensity at both synaptic regions and axonal regions. Each image gives a ratio of fluorescence intensity between synapses and axons. n refers to the number of images used for quantification.

## Acknowledgements

We would like to thank Yuji Kohara for kindly providing *apb-1* cDNA clone. We also thank Christian Frøkjær-Jensen for providing MosSCI vectors, Rob Hobson for *Punc-47::SNB-1::tagRFP* vector, Yueqi Wang for strain maintenance and Kim Schuske for suggestions on the manuscript. EM Jorgensen is an Investigator of the Howard Hughes Medical Institute.

## Additional information

### Funding

| Funder | Grant reference number | Author |
| --- | --- | --- |
| National Institutes of Health | NS034307 | Erik M Jorgensen |
| National Institutes of Health | GM067237 | Barth D Grant |

The funder had a role in study design for this work.

### Author contributions

MG, Conception and design, Acquisition of data, Analysis and interpretation of data, Drafting or revising the article; QL, SW, Acquisition of data, Analysis and interpretation of data; LS, Acquisition of data, Analysis and interpretation of data, Drafting or revising the article; GH, Drafting or revising the article, Contributed unpublished essential data or reagents; BDG, Conception and design, Analysis and interpretation of data, Drafting or revising the article, Contributed unpublished essential data or reagents; EMJ, Conception and design, Analysis and interpretation of data, Drafting or revising the article

## Additional files

### Supplementary files

• Supplementary file 1. Strains.

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
