## [Decision Letter]

Thank you for choosing to send your work entitled “AP2 subunits contribute independently
to synaptic vesicle endocytosis” for consideration at *eLife*. Your
article has been favorably evaluated by a Senior editor and 3 reviewers, one of whom is
a member of *eLife's* Board of Reviewing Editors. The following
individuals responsible for the peer review of your submission want to reveal their
identity: Graeme Davis (Reviewing editor); Tim Ryan (peer reviewer).

The Reviewing editor and the other reviewers discussed their comments before we reached
this decision, and the Reviewing editor has assembled the following comments based on
the reviewers' reports.

All three of the reviewers agree that the data set is strong, particularly the diversity
of approaches and strong genetic data used to demonstrate that APA-2 and APM-2 are both
required for endocytosis in *C. elegans*. All agree that the data will
appeal to a wide audience. Two reviewers are strongly supportive of the manuscript and
suggest minor textual modifications. A third reviewer has suggested additional
experiments and textual revisions. The additional experiments that were suggested would
directly test the hemi-complex model using labeled cargos specific for each
hemi-complex. While these experiments would significantly strengthen the manuscript,
they are not required for publication. When revising your manuscript, please address the
textual revisions suggested by each reviewer, listed below.

**Suggested experiments to improve the manuscript:**

One could imagine that different endocytic cargos could have differential sensitivity to
AP2 perturbations. An experiment would be to test cargos that are solely trafficked by
either hemi-complex. This could be done using chimeric CD4 proteins containing either
di-leucine (as in Figure 4) or tyrosine-based endocytic motifs (which mediate binding to
alpha2 and mu2 respectively). If hemi-complexes function independently, di-leucine
substrate recycling would be disrupted in apa-2 (as indicated in Figure 4D) but
unaffected in apm-2 (which was not tested). The converse pattern would be expected for
cargos containing a tyrosine endocytic motif. In principle, these experiments could
persuasively show that in mutants lacking specific subunits, the remaining AP2 subunits
have the capacity to function as hemi-complexes.

**Required textual revisions:**

1) Please revise the text to make it as clear as possible how much residual AP2 function
may exist in the hemi knockouts.

2) Please discuss the contribution of hemi-complex function to the wild type animal,
when all subunits are present. For example, AP2 binding to cargo endocytic motifs is
activated by PIP2 binding, which is mediated by residues in both hemi-complexes.
Clathrin recruitment by AP2 is mediated by clathrin binding sequences in both
hemi-complexes. Native cargoes may contain motifs that bind both alpha and mu subunits.
For these reasons, the avidity of the AP2 holo-complex for substrates and clathrin may
be much stronger than for either hemi-complex. In addition, the authors should comment
on the observation that protein abundance for the residual hemi-complex is relatively
low (e.g., in the *apm-2* mutants), suggesting that it would be a minor
contributor to wild-type endocytic activity. This is particularly the case if
recruitment to the membrane is compromised for the hemi-complex.

3) Please revise the text to include p-values for all data upon which arguments or
conclusions are based. This is particularly important for the diverse double mutant
phenotypes that appear to trend toward significance, but which may or may not reach
statistical significance. This is also the case when discussing comparisons being made
regarding electrophysiological data.

4) It is unclear why the authors did not report on the electrophysiological signature of
the double (alpha/mu) mutant in the “skin” rescued animals. This is worth at least
mentioning.

5) The authors should reference the mammalian work showing that loss of mu2 leads to
loss of alpha and point out that this is in contrast to the findings in this system.

6) It is not possible, at the light level, to know whether an altered axon/synapse ratio
of protein distribution has anything to do with the recycling of individual proteins.
This should be discussed.

7) The changes in vesicle size and number are similar to that observed in synaptotagmin
1 mutants and following acute inactivation of synaptotagmin 1. This is worth citing
since synaptic vesicle endocytosis is never directly assayed in the current study.

8) Figure 8C: “These data suggest that the AP2 complex plays a role in regulating the
size of synaptic vesicles.” Does the defect in vesicle size imply a role for determining
the size of the vesicle? Many endocytic mutations alter vesicle size and it is unlikely
that all participate in the control of vesicle diameter. An alternative conclusion is
that any mutation that disrupts the fidelity of endocytosis may well disrupt the ability
to reliably generate small vesicles of a consistent size.

---

## [Author Response]

***Suggested experiments to improve the manuscript*:**

*One could imagine that different endocytic cargos could have differential
sensitivity to AP2 perturbations. An experiment would be to test cargos that are
solely trafficked by either hemi-complex. This could be done using chimeric CD4
proteins containing either di-leucine (as in Figure 4) or tyrosine-based endocytic
motifs (which mediate binding to alpha2 and mu2 respectively). If hemi-complexes
function independently, di-leucine substrate recycling would be disrupted in apa-2
(as indicated in Figure 4D) but unaffected in apm-2 (which was not tested). The
converse pattern would be expected for cargos containing a tyrosine endocytic motif.
In principle, these experiments could persuasively show that in mutants lacking
specific subunits, the remaining AP2 subunits have the capacity to function as
hemi-complexes*.

As suggested by the reviewers, we engineered CD4 with the di-leucine α-adaptin
recognition motif and tested its localization in α or μ2 mutants. Consistent with the
model, the substrate was strongly mislocalized in the *apa-2* α mutant,
and weakly mislocalized in the *apm-2* μ2 mutant. Images are shown in
Figure 5D and the quantification is shown in Figure 5E. For a μ2 substrate, we used
MIG-14 / wntless, which we have previously shown requires μ2 for endocytosis from the
surface. In reciprocal experiments we found that MIG-14 endocytosis is more defective in
μ2 mutants than α mutants. Images are in Figure 5F and quantification in Figure 5G.

***Required textual revisions*:**

*1) Please revise the text to make it as clear as possible how much residual AP2
function may exist in the hemi knockouts.* In response to this request, we
added two experiments to describe residual AP2 function in the mutants. First, we
quantified the levels of remaining AP2 subunits by western blot in each of the mutants
(depending on the availability of functional antibodies; Figure 6—figure supplements
1&2). Second, we quantified the fluorescence intensity of all four AP2 subunits at
the nerve ring in α and μ2 mutants Figure 6 and Table 1).

*2) Please discuss the contribution of hemi-complex function to the wild type
animal, when all subunits are present. For example, AP2 binding to cargo endocytic
motifs is activated by PIP2 binding, which is mediated by residues in both
hemi-complexes. Clathrin recruitment by AP2 is mediated by clathrin binding sequences
in both hemi-complexes. Native cargoes may contain motifs that bind both alpha and mu
subunits. For these reasons, the avidity of the AP2 holo-complex for substrates and
clathrin may be much stronger than for either hemi-complex. In addition, the authors
should comment on the observation that protein abundance for the residual
hemi-complex is relatively low (e.g., in the apm-2 mutants), suggesting that it would
be a minor contributor to wild-type endocytic activity. This is particularly the case
if recruitment to the membrane is compromised for the hemi-complex*.

It is true, the holocomplex is likely to provide some inseparable functions, and even if
completely stable, the function of a hemicomplex might be compromised. It is also
important to note that the hemicomplexes are *not* fully stable. We now
emphasize these two points in the new version of the Discussion.

*3) Please revise the text to include p-values for all data upon which arguments
or conclusions are based. This is particularly important for the diverse double
mutant phenotypes that appear to trend toward significance, but which may or may not
reach statistical significance. This is also the case when discussing comparisons
being made regarding electrophysiological data*.

We double-checked our comparisons and have included all p-values in the legends of the
figures or the figure supplements. To address the electrophysiological comparisons more
precisely, we added the following text:

“In *apa-2(ox422)* mutants, the amplitude from miniature spontaneously
released vesicles (minis) is increased by 40% (Figure 9C). The mini amplitudes in the
skin-rescued single and double mutants are also larger, although they do not reach
statistical significance.”

*4) It is unclear why the authors did not report on the electrophysiological
signature of the double (alpha/mu) mutant in the “skin” rescued animals. This is
worth at least mentioning*.

The data for the skin-rescued double mutants are included in Figure 9. In the text we
report the results as follows: “The double mutants exhibit a more severe, 42% reduction
in the amplitude of the evoked responses (Figure 9E).”

“There is also a more severe reduction in the rates of tonic synaptic vesicle fusion.
Skin-rescued *apa-2* animals exhibit a 50% reduction in mini frequency,
and the skin-rescued *apa-2 apm-2* double mutants exhibit a 68% reduction
in mini frequency (Figure 9D).”

*5) The authors should reference the mammalian work showing that loss of mu2
leads to loss of alpha and point out that this is in contrast to the findings in this
system*.

We now note this difference in the Discussion.

*6) It is not possible, at the light level, to know whether an altered
axon/synapse ratio of protein distribution has anything to do with the recycling of
individual proteins. This should be discussed.* In *C.
elegans,* stonin / UNC-41 (Mullen et al., 2012) and AP180 / UNC-11 (Nonet et
al., 1999) are two characterized adaptors for synaptotagmin and synaptobrevin recycling
at synapses. In the absence of these adaptors, the tagged cargo protein is severely
diffused into axons that make the fluorescence at the synaptic varicosities decrease or
even out with that in axons. So we still believe that the axon/synapse ratio of protein
distribution could reflect the efficiency of the cargo recycling to some extent. To make
this point clear in the paper, we have made the following change:

“In *C. elegans* mutants lacking particular adaptor proteins, the cognate
cargo protein diffuses into axons. For example in AP180 mutants, synaptobrevin is no
longer concentrated at synapses but is diffuse in axons (Nonet et al., 1999). By
contrast, in AP2 adaptin mutants, synaptic vesicle proteins are not grossly
mislocalized.”

*7) The changes in vesicle size and number are similar to that observed in
synaptotagmin 1 mutants and following acute inactivation of synaptotagmin 1. This is
worth citing since synaptic vesicle endocytosis is never directly assayed in the
current study*.

That is correct: acute inactivation of synaptotagmin at the *Drosophila*
neuromuscular junction as well as chronic inactivation in *C. elegans*
exhibits a similar phenotype as that found here for AP2 knockouts. Moreover, we
previously found that stonin shares overlapping functions with AP2. We now discuss the
relationship among synaptotagmin, AP2 and stoning in the Discussion.

*8) Figure 8C: “These data suggest that the AP2 complex plays a role in
regulating the size of synaptic vesicles.” Does the defect in vesicle size imply a
role for determining the size of the vesicle? Many endocytic mutations alter vesicle
size and it is unlikely that all participate in the control of vesicle diameter. An
alternative conclusion is that any mutation that disrupts the fidelity of endocytosis
may well disrupt the ability to reliably generate small vesicles of a consistent
size*.

We have amended the text to state that the change in vesicle diameter might be indirect.
In addition, to try to pinpoint where the defect in AP2 mutants might lie, we analyzed
more sections and performed a 3D reconstruction of a synapse from the *apa-2
apm-2* double mutant. Defects are observed at the cell membrane in some
cases, but the most prominent defects are large vacuoles present in the synaptic
varicosity. We speculate that AP2 might have a late role in the regeneration of synaptic
vesicles from endosomes. The changes are as follows: Results, “These data suggest that
the AP2 complex may play a role in regulating the size of synaptic vesicles.
Alternatively, the effect on vesicle size may be indirect due to pleiotropic defects in
endocytosis.”

Discussion, “The most prominent defect observed in AP2 mutants is the presence of large
diameter vesicles and vacuoles in the synapses of the α-μ2 adaptin double mutants. […]
These data suggest that AP2 has a late function and possibly AP2 is required to
regenerate synaptic vesicles from endosomes.”